# Microstructure, Mechanical Properties and Corrosion Behaviors of Al–Li–Cu–Mg–Ag–Zn Alloys

**DOI:** 10.3390/ma15020443

**Published:** 2022-01-07

**Authors:** Mingdong Wu, Daihong Xiao, Xinkai Wang, Lanping Huang, Wensheng Liu

**Affiliations:** Science and Technology on High-Strength Structural Materials Laboratory, Central South University, Changsha 410083, China; mmingshiddong@csu.edu.cn (M.W.); wangxinkai@csu.edu.cn (X.W.); christie@csu.edu.cn (L.H.)

**Keywords:** aluminum lithium alloys, Zn addition, mechanical properties, corrosion resistance, microstructure

## Abstract

Combined with microstructure characterization and properties tests, the effects of Zn contents on the mechanical properties, corrosion behaviors, and microstructural evolution of extruded Al–Li–Cu–Mg–Ag alloys were investigated. The results show that the increase in Zn contents can accelerate hardening kinetics and improve the hardness of peak-aged alloys. The Zn-added alloys present non-recrystallization characteristics combined with partially small recrystallized grains along the grain boundaries, while the T1 phase with finer dimension and higher number density could explain the constantly increasing tensile strength. In addition, increasing Zn contents led to a lower corrosion current density and a shallower maximum intergranular corrosion depth, thus improving the corrosion resistance of the alloys. Zn addition, distributed in the central layer of T1 phases, not only facilitates the precipitation of more T1 phases but also reduces the corrosion potential difference between the T1 phase and the matrix. Therefore, adding 0.57 wt.% Zn to the alloy has an excellent combination of tensile strength and corrosion resistance. The properties induced by Zn under the T8 temper (solid solution treatment + water quenching + 5% pre-strain+ isothermal aging), however, are not as apparent as the T6 temper (solid solution treatment + water quenching + isothermal aging).

## 1. Introduction

Due to lower density, better specific strength, higher specific stiffness, and higher fatigue performance compared with conventional Al alloys [1], Al–Li alloys are appealing prospects for critical structures used in the aerospace and aircraft industries [2,3]. Li added to Al alloys simultaneously reduces the density and the stiffness of the alloys, while combining it with Cu enhances precipitation strengthening, resulting in a high level of strength [4]. However, Li addition promotes the formation of anodic precipitation phases such as T1 (Al_2_CuLi) and T2 (Al_6_CuLi_3_) at the grain boundaries, which causes the alloys to be more prone to intergranular corrosion (IGC) [5]. IGC has an adverse impact on the safety, dependability, and service life of aircraft structural materials. As a result, the coordination between strength and corrosion behavior of Al-Li alloys has attracted increasing attention.

Alloying has been the mainly used strategy to improve the overall performance of Al–Li alloys. The performance of Al–Li–Cu alloys is primarily affected by precipitates, including T1, θ’(Al_2_Cu), δ’(Al_3_Li), and S’(Al_2_CuMg) [6]. However, precipitates are largely related to the added alloying elements [7]. The types and volume fractions of precipitations in Al–Li–Cu alloys depend on the concentration of Cu and Li and Cu/Li ratio [8,9]. Furthermore, the addition of microalloying elements diversifies and complexities the shape and kinds of precipitates. Mg addition, for example, has been shown to enhance hardness by reducing Li solubility in the matrix to promote a more fine and uniform distribution of T1 phases [10,11]. The combined addition of Mg and noble metal Ag that distributes in the interior of the T1 phase also increases the properties of the alloy by promoting the precipitation kinetics of the T1 phase and increasing its nucleation efficiency [12].

In parallel, Zn is added to Al–Li–Cu–Mg–Ag alloys as a microalloying element in combination with Mg to further increase the properties of the alloys, in which the T1 phase dominates the precipitate. To better understand the role of Zn in the precipitation of T1 phase, Grumman et al. [13,14] studied the composition of the T1 phase under different aging times. The results suggested that Zn was observable in the T1 phase and was more likely to replace Cu atoms. According to Kertz et al. [15], Cu occupied by Zn in the T1 phase may reduce the electrochemical difference between the T1 phase and the matrix, which could increase the electrochemical potential of the intragranular zone of the aged alloys. By studying the corrosion performance of Zn on Al-4.0Li-1.7Cu-0.4Mg alloys (at.%), Liu and the coworkers [16] also found that the precipitates containing Zn at the grain boundary may reduce potential differences between grain boundary and precipitation free zone (PFZ). In addition, coarsened and discontinued T1 phases at the grain boundary for the increase in Zn concentration could destroy corrosion channels along the boundary [17], thus improving the corrosion performance of alloys. Studies above demonstrated that Zn elements have a positive effect on corrosion behaviors. Besides, Zn addition can also improve the mechanical properties of Al–Li–Cu alloys [17,18]. Sun et al. [19] discovered that Zn efficiently enhanced the precipitation of the T1 phase while inhibited the formation of the θ’ phase in the matrix. Their findings indicated that the tensile properties of Al-1Li-4Cu-0.4Mg-0.1Zr-xZn alloys (wt.%) gradually improved with the increase in Zn contents.

To sum up, researches have been conducted into the influence of Zn additions on the evolution of microstructure and associated properties in Al–Li–Cu–Mg–Ag alloys during aging treatment. There is still a lack of knowledge for the understanding of Zn in precipitation and texture. The principal purpose of this study was to investigate the effect of Zn contents on the mechanical performance, corrosion behavior, and microstructural evolution of new Al–Li alloys under various aging conditions, and then to balance mechanical and corrosion properties in Al–Li–Cu–Mg–Ag alloys. The mechanism of Zn on crystal structure and the electrochemical behavior of the T1 phase were discussed.

## 2. Experimental Details

### 2.1. Specimen Preparation and Heat Treatment

The Al–Li–Cu–Mg–Ag-Zn alloys in this work were prepared by melting and casting. Commercial pure Al, pure Mg, pure Zn, pure Ag, pure Li, and master alloys of Al-50Cu, Al-10Mn, and Al-5Zr were placed in a vacuum resistance melting furnace (ZGG-0.025A, Shanghai, China) at 750 °C and then poured into a cylindrical mold with a diameter of 100 mm. The chemical composition of the prepared alloys was measured by inductively coupled plasma atomic emission spectrometry (ICP-AES, iCAP7600, Thermo Fisher Scientific, Waltham, MA, USA). The actual compositions of the studied alloys are listed in Table 1. The as-cast ingots were subjected to continuous three-stage homogenization treatment (400 °C × 10 h + 450 °C × 10 h + 500 °C × 24 h) to ensure the majority of the secondary-phase constituents dissolved in the Al matrix. Subsequently, the homogenized ingots were hot extruded into plates when heated at 450 °C for 80 min; the extrusion ratio of plates is 9. After the extrusion process, the extruded plates were immediately cooled by water. The alloys were solution-treated at 530 °C for 1 h in molten salt and then quenched in water at room temperature. To study the impact on the corrosion behaviors and mechanical properties under different aging conditions, the plate samples were divided into two groups (T6 and T8 temper). Before aging at 160 °C for various durations, the T8 treatment samples, unlike T6 treatment without being pre-deformed, were immediately plastic deformation through cold rolling to a degree of 5%.

### 2.2. Hardness and Tensile Testing

The hardness was tested as a function of the aging time using an HVS-50 microhardness instrument (Huayin, Laizhou, China) with a load of 3 kgf and a dwelling time of 15 s. Hardness was measured every 5 h for the first 40 h, then every 10 h. The hardness value of each sample was taken as the mean of seven measurements. An Instron 3369 (Instron, Boston, MA, USA) universal test machine equipped with a contact extensometer was employed to assess the room temperature tensile performance at a displacement rate of 2 mm/min. The tensile test samples, a cross-section of 6 mm (width) × 2 mm (thickness) and a gauge length of 30 mm along the sheet extrusion direction (ED), were prepared using a wire cutting machine. The tensile results of each condition were calculated on three measurements.

### 2.3. Corrosion and Electrochemical Testing

IGC tests and polarization measurements were carried out to investigate the corrosion behavior of the alloys. The samples for intergranular corrosion tests were firstly polished with 1.5 μm Al_2_O_3_, then cleaned with ethanol, and finally dried by electric breeze. The IGC test was immersing the samples in a solution of 57 g/L NaCl + 10 mL/L H_2_O_2_ at 35 ± 2 °C for 6 h. Optical microscopy (OM, Leica DM4500P, Wizz, Germany) was used to characterize the corrosion damage from the cross-section of the sample that was perpendicular to the rolling direction. Open circuit potential and polarization tests were conducted with a CHI660C (Shanghai Chenhua, Shanghai, China) electrochemical instrument in a 3.5 wt.% NaCl solution, where the corrosion potential was tested at a scanning rate of 1 mV/s. The platinum, the saturated calomel electrode (SCE), and the sample were used as the auxiliary electrode, the reference electrode, and the working electrode, respectively.

### 2.4. Microstructural Analysis

The observation position of all specimens was taken from the central layer of the extruded plates. OM was used to observe the microstructure of the samples after grinding, polishing, and finally corroding with Keller’s reagent (1% HF + 1.5% HCl + 2.5% HNO_3_ + 95% H_2_O, vol.%). Electron backscatter diffraction (EBSD) measurement was performed by a scanning electron microscope (SEM, ZEISS EVO MA10, Zeiss, Oberkochen, Germany) equipped with an EBSD (NordlysMax3, Oxford instrument, Oxford, UK) detector. After mechanical polishing, specimens were electro-polished using a solution (10% HClO_4_ + 90% C_2_H_5_OH, vol.%) at 20 V to generate higher quality EBSD maps. Samples with the dimension of 0.5 × 0.5 × 15 mm^3^ for an atom probe tomography (APT) study were obtained by electro-discharge machining, and then prepared by electro-polishing techniques using a solution (33% HNO_3_ + 67% CH₃OH, vol.%) at 14 V. APT analysis was carried out on a LEAP 4000 HR (CAMECA, Gennevilliers, France) instrument at a pulse fraction rate of 200 kHz and pulse fraction of 20%, and the specimens were held at −253 °C. Data reconstruction and three-dimensional (3D) visualization were obtained from the APT data by IVAS 3.6.12 software (CAMECA, Gennevilliers, France). Transmission electron microscope (TEM, JEM-2100F, JEOL, Tokyo, Japan) was applied to characterize the density number, size, and distribution of precipitates. The average diameter of T1 phases was calculated by manually measuring the size of T1 plates from at least three pictures using the Image J analysis software (1.8.0, 2021, National Institutes of Health, Bethesda, MD, USA). Specimens prepared for TEM observation were firstly mechanical ground to 70 μm in thickness, and then electropolished by twin-jet electrolytic thinning instrument with a mixed solution of 25% HNO_3_ and 75% CH_3_OH at −30 °C.

## 3. Results

### 3.1. Aging Hardening Behaviors

Figure 1 shows the isothermal age-hardening curves of the extruded alloys aged at 160 °C in T6 and T8 tempers. The Zn-free alloy 1 in the T6 treatment reaches a peak hardness of 189 HV after aging 50 h, while Zn-added alloys 2 and 3 only need aging for 40 h and 35 h, respectively. The corresponding harnesses are 196 HV and 202 HV, indicating that the increase in Zn contents can accelerate hardening kinetics and improve hardness value. Compared to the T6 state, pre-deformation after the solid solution in Figure 1b presents a significant age-hardening response, practically from the initial quenched deformation to the aging time of 5 h. Alloys 1, 2, and 3 in the T8 state reach the peak aging hardness of 191 HV (35 h), 192 HV (35 h), and 201 HV (20 h), respectively. The hardness values of the alloys follow downward trends with the extension of aging time after attaining the peak value. Although the hardness of alloy 3 containing 0.57 wt.% Zn evidently decreases following the peak aging, it remains unchanged after reaching the plateau.

### 3.2. Tensile Properties

Figure 2a,b present the tensile properties of the alloys after T6 treatments at 160 °C for 32 h and 80 h, respectively. It is apparent from these figures that increasing the Zn contents further improves the tensile strength (UTS). Compared with the alloy 1 aged for 32 h, the yield strength (YS) of alloys 2 and 3 increased by 24 MPa and 70 MPa, respectively. For the alloys aged for 80 h, the Zn-containing alloys show higher YS and UTS strength, while the elongation (El) slightly declines. The YS and UTS of T8 treated alloys are higher than those of T6 treated alloys, as shown in Figure 2c,d. However, the improvement of strength by Zn addition in the T8 state, almost 615 MPa, is not significant compared to the T6 state (Figure 2c). The YS and UTS of all alloys are reduced as the aging time prolongs to 80 h. The alloy 3 aged at 160 °C for 80 h under T8 treatment has an excellent mix of strength (YS = 622 MPa, UTS = 651 MPa) and elongation (EL = 6.4%). Interestingly, the data in this condition show that the change in YS was not as noticeable when treated in T8 tempers.

### 3.3. Corrosion Behaviors

Figure 3 shows the IGC morphology and maximum corrosion depth of the alloys aged at 160 °C for 32 h in the T6 and T8 states. All samples present local corrosion characteristics. In the T6 state, the increase in Zn contents reduces the maximum IGC depth. The maximum IGC depth corresponding to alloys 1, 2, and 3 is 77.31 μm, 53.03 μm, and 46.15 μm, respectively. Under T8 conditions, alloys with different Zn contents have a similar trend to T6 treatment, whereas the maximum IGC depth decreased with the increase in Zn contents.

The polarization curves and their related electrochemical parameters of the alloys after the T6 and T8 treatment for 32 h are shown in Figure 4 and Table 2, respectively. Under the T6 treatment, the corrosion potential of the alloys decreases as increasing Zn contents. Plastic deformation prior to the aging process makes the corrosion potential become more negative compared with T6 treated samples. The corrosion current that reflects the rate of corrosion gradually decreases with the gradual increase in Zn contents. Under the T6 treatment, the corrosion current of alloys 1 and 2 are 3.10 × 10^−5^ A/cm^2^ and 3.17 × 10^−6^ A/cm^2^, respectively, which is higher than the alloy 3 (1.81 × 10^−6^ A/cm^2^). In the T8 state, the corrosion current of the alloys is further reduced and shows the same trend as the T6 temper. These results suggest that Zn addition could improve the corrosion resistance of the alloys.

### 3.4. Microstructure

OM images of the extruded alloys after solid solution treatment are presented in Figure 5. Obviously, the grain shape of the extruded alloys after solution treatment is an elongated state that parallels the ED. There are obvious differences between the grain characteristic of the two alloys. The fibrous grains of the alloys become coarser and more continuous after adding Zn elements. Fine equiaxed grains produce along with the fiber-like structures, indicating that non-recrystallization features coupled with partial recrystallization are present.

To obtain more detailed information on grain orientation and spatial distribution, orientation image microscopy maps and a misorientation angle distribution histogram along the ED-TD plane of samples are illustrated in Figure 6. It is interesting to note that, as shown in Figure 6d–f, the proportion of small-angle grain boundaries in the alloys first decreases and then increases with the increase in Zn contents. The proportions of low-angle grain boundaries in alloys 1, 2, and 3 were 74.7%, 73.60%, and 85.3%, respectively. Apparently, the grain orientation of extruded alloys is parallel to the ED fiber texture <111> and <100>, which is the typical behavior of the uniaxial deformation process of Al alloys. The scattered and contoured pole figure along the ED, as shown in Figure 7, can further indicate this phenomenon. All alloys present strong <111> and <001> orientation aggregation under the same extrusion process after solid solution treatment. The strong <111> and weak <001> direction fibrous structures present in the alloy 2 (Figure 7b). In the Zn-free alloy 1 (Figure 7a), however, the main fiber structure is in the <001> direction, and there is also a fiber structure between <111> and <001>. The alloy 3 with 0.57 wt.% Zn content is dominated by fiber structure between <111> and <001> directions (Figure 7c).

Figure 8 shows the bright field (BF) images and the corresponding selected area electron diffraction (SAED)patterns taken along the [110]_Al_ zone axis from alloys 1 and 3 after aging at 160 °C for 32 h. Compared with the BF images of the alloy 1 in the T6 conditions (Figure 8a,d), the finer size and higher number density of T1 phases are precipitated due to increasing the Zn content to 0.57 wt.% (alloy 3), which can be further confirmed by the corresponding SAED patterns. Under T8 treatment, the BF images in Figure 8b,e reveal that the pre-deformation before aging increases the number density of the T1 phase and shortens its diameter. In addition, the number density of T1 phases in the alloy 3 is much higher than the alloy 1. Figure 8c,f present the intergranular precipitation characteristics of alloys 1 and 3 after aging at 160 °C for 32h under T6 treatment, respectively. The alloy 1 exhibits continuous precipitates at grain boundaries, and the PFZ is formed nearby the coarsen intergranular precipitates. However, adding 0.57 wt.% Zn element has discrete and finer grain boundary precipitates, in which the width of PFZ is smaller than that of the Zn-free alloy 1.

Figure 9 and Figure 10 show the BF images taken along the [112]_Al_ zone axis and its corresponding length distribution diagrams of the T1 phase from alloys 1 and 3 after aging at 160 °C in the T8 temper, respectively. Only one of the four T1 phase variations is detected along the [112]_Al_ direction because of the orientation relationship between the T1 phase and the matrix. Similar to the results in Figure 8, there are a small number of irregularly distributed T1 phases with an average length of 53.4 nm in the alloy 1. The number density of the T1 phase in the alloy 3 increases under the same treatment conditions, and the average length of the uniform distributed the T1 phase is 48.7 nm (Figure 9c and Figure 10c). When aging for 80 h, the number density and visible growth of the T1 phase in the alloy 1 increases and occurs, and the average length of the T1 phase is around 76.8 nm (Figure 9b and Figure 10b). Compared with the Zn-free alloy 1, a large number of fine T1 precipitates are present in the alloy 3. Therefore, adding Zn to the alloys can enhance the precipitation density of the T1 phase and refine it. The length of the T1 phase in the Zn-containing alloys is shorter, although it further elongates during long-term aging treatment.

Figure 11 displays the TEM images of grain boundaries from alloys 1 and 3 in the T8 state. There is no obvious PFZ nearby the grain boundaries after pre-deformation. Coarsened and dispersed precipitates adjacent to grain boundaries have been found in the alloy 1 aged for 32 h (Figure 11a). With increasing Zn contents to 0.57 wt.% (alloy 3), a large number of refined and discontinued precipitates decorate the boundary in the same condition (Figure 11b); moreover, the precipitates coarsen as the aging time reached 80 h (Figure 11c).

To confirm the distribution of Zn atoms in the plate-like T1 phase, the alloy 3 was examined in atom probe tomography after aging at 160 °C for 32 h, as shown in Figure 12. As illustrated in Figure 12a, there are many plate-like precipitates, marked by the orange 11 at.% Cu iso-concentration surfaces, in the APT volume, and cross-interacted platelets with an angle of about 70° are confirmed, corresponding to the angle between the two {111}_Al_ planes. The composition evolution across the plate-like dispersoids in Figure 12b demonstrates that the precipitate is mainly Cu and Li elements while enriching Mg, Zn, and Ag atoms. These results support the ideas of Deng [20] and Gao [21], who confirmed that the precipitate was the T1 phase. It can be seen from the data in Figure 12c that Zn has a similar distribution characteristic with Ag and Mg in the T1 phase. Proximity histograms also confirm the hypothesis [18] that the density of Zn elements in the core of the T1 particle is higher than that separated at the T1/matrix interfaces.

## 4. Discussion

### 4.1. Effect of Zn Contents on Aging Precipitation

Many investigations have found Zn in both thick T1 precipitates and thicker ones [18]. Zn is not predicted to integrate into other phases in competition with T1, such as θ’, which appears to be helpful in the formation of the T1 phase. The T1 phase is a four-layer crystal structure, and each of its units comprises a Li-rich central layer between two corrugated Cu-rich layers with the plate bounded by Al–Cu layers [22]. The APT images, as shown in Figure 12, prove that Zn atoms enrich in the precipitate. It can be proposed that the density of Zn at the central of the T1 particle is higher than that in the edge area. This phenomenon also suggests that Zn atoms access into the T1 phase at the initial stage of aging. According to the results of Niessen et al. [23], the binding ability of Zn between Cu and Li is relatively strong. Thus, the probability of Cu and Li atoms being trapped by Zn atoms enhances continuously with the increase in Zn contents. In addition, Zn atoms could reduce the solution of Li in the matrix [24]. Therefore, Zn addition facilitates the precipitation of T1 phases containing Zn, Cu, and Li in the aging process.

The segregation of solute atoms such as Ag, Mg on the {111}_Al_ plane could decrease the stacking fault energy of Al alloys [25]. Many studies have confirmed that Zn had a similar effect as Ag because both Zn and Ag combined additions with Mg could effectively promote the precipitation of the T1 phase [13,14]. Zn atoms have a high solubility in the Al matrix, and they may also segregate at the {111}_Al_ plane. Therefore, Zn addition is beneficial to the formation of stacking faults after plastic deformation. It is generally believed that stacking faults can be regarded as the nucleation site of the T1 phase [26]. The addition of Zn in Al alloy not only leads to the generation of dislocations during pre-deformation prior to the aging but maintains numerous deformed structures (Figure 7). Therefore, adding Zn to the Al–Li–Cu–Mg–Ag alloy can promote the number density of the T1 phase [19]. As aging prolongs, however, Zn elements should migrate into the matrix due to the growth and coarsening of the T1 phase. The migration of Zn atoms in this process requires higher activity energy [12,23]. It can thus be suggested that Zn addition not only promotes the nucleation of the T1 phase but inhibits its growth.

### 4.2. Effect of Zn Contents on the Mechanical Properties

The contoured pole figure demonstrates a weaker <001> fiber texture along with <111> fiber texture in the extrude alloys, as shown in Figure 5. Fiber-like textures are inevitably formed in metal materials that undergo extrusion [27]. For extruded FCC alloy, the <100> texture is recognized as the recrystallized texture [28]. The recrystallization proceeds as the matrix rotates away from the original orientation towards <001> [29,30]. Zn addition suppresses the transformation of fiber textures from <111> to <001> to some extent. The microstructure containing recrystallized structures and those that have not been recrystallized can improve the plastic properties of the alloys. Therefore, the addition of Zn enables the alloy to obtain higher strength while maintaining excellent plasticity (Figure 2).

It is known that the mechanical properties of Al–Li–Cu alloys mainly depend on the precipitation characteristics in grain interiors. The strength contributed from the T1 phase, according to the equation proposed by Dorin et al. [31,32], can be described as:(1)$$\sigma_{P}\propto D^{2}N^{\frac{1}{2}}t^{-}{}^{\frac{3}{2}}$$
where Δ*τ* is the precipitation hardening provided by the T1 phase, *D*, *N*, and *t* are the average length, the number density, and the thickness of the T1 phase, respectively. When the thickness *t* of the precipitated T1 phase remains constant, the enhancement of the precipitation in the aging process only depends on the average diameter *D* and the number density *N* of the T1 phase. Therefore, the diameter of the T1 phase has a more significant influence on the strength. The alloy softens, however, mostly due to the thickening or coarsening of T1 precipitates.

Figure 1 shows that the addition of a small amount of Zn element slightly improves the hardness of Al–Li–Cu–Mg–Ag alloys. Furthermore, the YS and UTS of the alloys increase with the gradual rise in Zn contents, as shown in Figure 2. According to the TEM observation in Figure 7 and Figure 8, the contribution of the T1 phase on the strength of Al–Li–Cu alloys is characterized quantitatively using Equation (1). After T6 treatment, Zn addition to the alloys facilitates the formation of finer and more uniformly distributed T1 phases. Stress concentration could arise at these phases, resulting in an increase in strength after T6 treatment. Figure 8 suggests that the addition of Zn also increases the precipitation density of the T1 phase under T8 treatment. However, the improvement of strength by Zn addition in the T8 state is not significant compared to the T6 temper. The yield strength of the alloys is almost equal. Pre-deformation before the aging process causes the precipitation of the T1 phase more efficiently when compared with T6 samples. Therefore, the promotion of Zn to precipitating T1 phase is not as considerable as that of the T6 condition. As aging prolongs to 80 h in the T8 state, although the length of the T1 phase increases, the coarsened T1 phase lead to a decrease in strength, particularly in the Zn-free alloy 1. The presence of Zn in the T1 phase improves the coarsening resistance of the Zn-containing alloys, slowing the strength decline [19].

### 4.3. Effect of Zn Contents on Corrosion Performance

Figure 3 and Figure 4 indicate that Zn addition in the alloys could facilitate a lower maximum intergranular corrosion depth and a negative potential. Those results can be explained as follows. First, the T1 phase, which precipitates simultaneously at grain boundaries and in grain interiors, significantly impacts on the corrosion potential [33]. Proton et al. [34] suggested that the formation of the T1 phase within grains reduces the solute Cu atoms in the matrix, leading to a decrease in the open-circuit potential. Therefore, the open circuit potential of the alloys mostly depends on the precipitates in the grains. It can be seen from Figure 9 that Zn addition increases the number density of the T1 phase in the grains, and the potential difference between the grains and the grain boundaries becomes smaller, which thus reduces the corrosion sensitivity of the alloys [9]. Besides, the refined and discontinued precipitates in Figure 11 promote the corrosion potential of grain boundary to shift to a more positive value as the Zn contents increase. Last but not least, the corrosion performance is determined by grain energy storage [35]. During the corrosion process, the subgrain, which has a large number of dislocations, is more likely to reduce the corrosion that originated in the grain boundary, while the matrix became more susceptible to act as a sacrificial anode. Therefore, the higher the proportion of subgrain boundaries in the Zn-containing alloys (Figure 6), the better the corrosion resistance of the alloys.

As shown in Figure 12, Zn atoms distribute around the T1 phase. Zn may substitute Cu atoms in the T1 phase [13,14], which could influence the electrochemical characteristics of the T1 phase [16]. The potential of the T1 phase is –1.089 V, which is lower than that of pure Al [36]. Therefore, the T1 phase preferentially dissolves in the corrosive medium. During the corrosion process, the corrosion potential positively shifts because of the preferential dissolution of active Li. Thus, the corroded T1 phase transforms into a cathode relative to the matrix and accelerates the dissolution of the adjacent matrix in the latter stage [17]. However, as Zn atoms replace part of the Cu atoms in the T1 phase, the electrochemical potential of the T1 phase containing Zn elements was more positive than that of the regular T1 phase. Zn and Li dissolve preferentially during the corrosion process, leading to an enrichment of Cu atoms in the matrix and a more positive potential of the T1 phase. This phenomenon indicates that the potential difference between the T1 phase and the matrix can be reduced, improving the corrosion resistance of the alloys with increasing Zn contents.

## 5. Conclusions

Improving the mechanical properties and corrosion resistance of Al–Li–Cu alloys plays an important role in aircraft industries, and micro-alloying has been considered to be the most effective approach to achieve this goal. In this work, the effects of different Zn contents on the properties and microstructure of novel Al–Li–Cu–Mg–Ag alloys under two aging conditions (T6 and T8) were studied in detail. Some major conclusions can be derived from the experimental data, which are as follows:

(1) The distribution of Zn in the center of the T1 precipitate is higher than being segregated at the T1/matrix interfaces. The segregation of Zn on [111]_Al_ plane can reduce the stacking fault energy of Al–Li alloys, which is beneficial to the formation of the T1 phase.

(2) The addition of Zn can significantly increase the hardness value and the tensile strength of the alloy under T6 and T8 conditions. However, the improvement of strength by Zn addition in the T8 state is not significant compared to the T6 state. Adding Zn to the alloys results in a combination of recrystallized and non-recrystallized grains in the solution treated alloys, which contribute to greater strength with excellent plasticity. The alloy 3 aged at 160 °C for 80 h under T8 treatment has a good mix of strength (YS = 622 MPa, UTS = 651 MPa) and elongation (EL = 6.4%).

(3) With increasing Zn contents, the maximum intergranular corrosion depth decreases as well as the open circuit potential becoming negative, which means the corrosion resistance of the alloys increases correspondingly. This might be due to the fact that substituting Zn for Cu in the T1 phase changes the electrochemical properties of the T1 phase and reduces the electrochemical difference between the T1 phase and the matrix.

## Figures and Tables

**Figure 1 materials-15-00443-f001:**
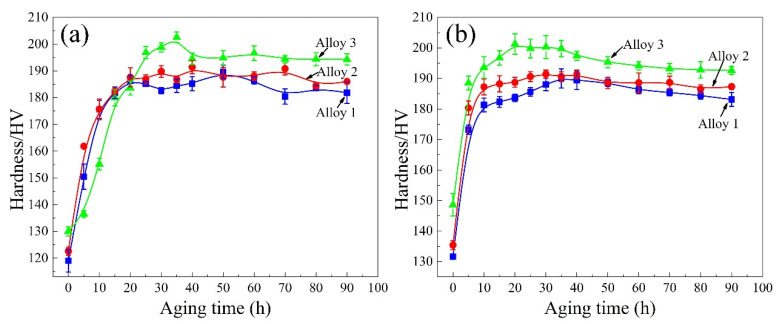
Age hardening curves of alloys after T6 (**a**) and T8 (**b**) treatments at 160 °C.

**Figure 2 materials-15-00443-f002:**
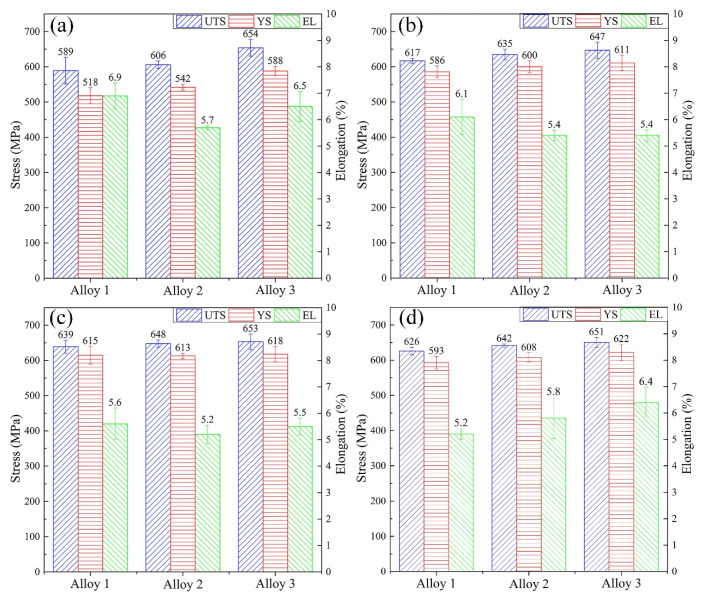
Tensile properties of alloys aged at 160 °C: (**a**) T6-32 h; (**b**) T6-80 h; (**c**) T8-32 h; (**d**) T8-80 h.

**Figure 3 materials-15-00443-f003:**
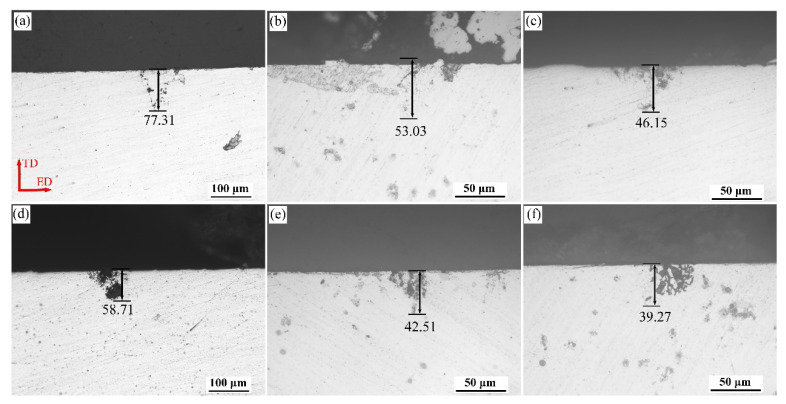
Optical microscopy (OM) micrographs of ED-TD (transverse direction) plane (perpendicular to the extrusion direction) of alloys 1 (**a**,**d**), 2 (**b**,**e**), and 3 (**c**,**f**) aged at 160 °C for 32 h in T6 (**a**–**c**) and T8 (**d**–**f**) states after immersion in an intergranular corrosion (IGC) environment for 24 h.

**Figure 4 materials-15-00443-f004:**
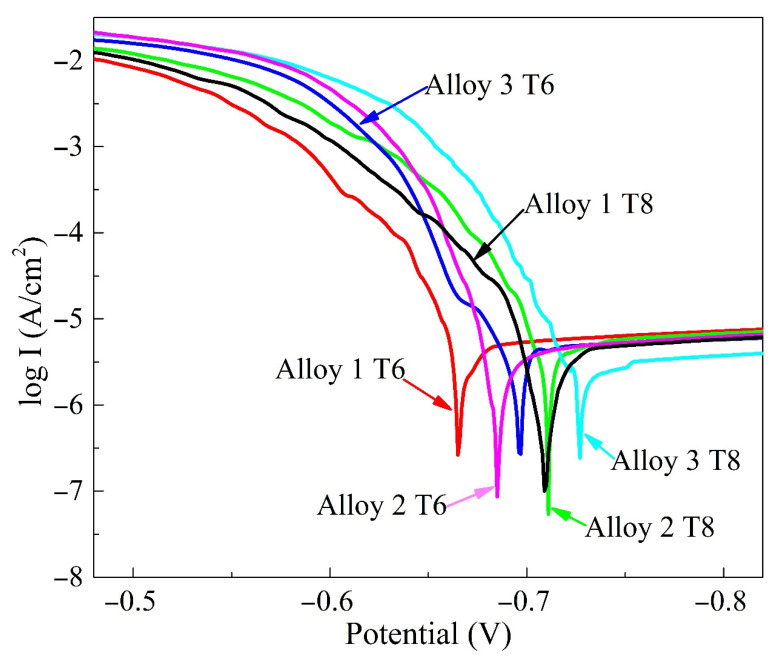
Tafel curves of alloys with different Zn content under different aging conditions.

**Figure 5 materials-15-00443-f005:**
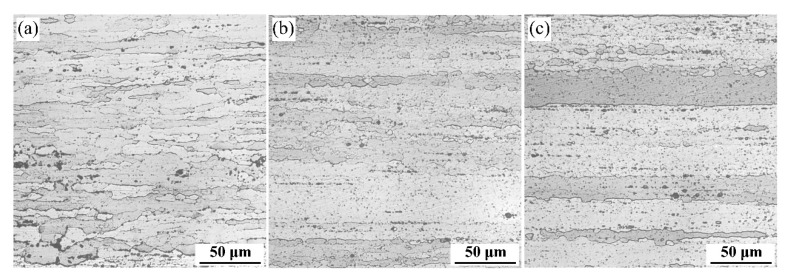
Optical microscopy (OM) images showing the microstructure of extruded alloys 1 (**a**), 2 (**b**), and 3 (**c**) after aging solution treatment.

**Figure 6 materials-15-00443-f006:**
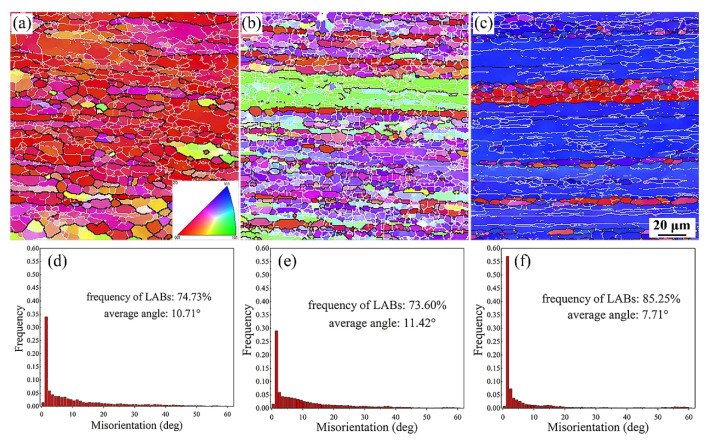
Electron backscatter diffraction (EBSD) maps of orientation distribution and the grain boundary misorientation statistical images of extruded alloys 1 (**a**,**d**), 2 (**b**,**e**), and 3 (**c**,**f**) after aging treatment.

**Figure 7 materials-15-00443-f007:**
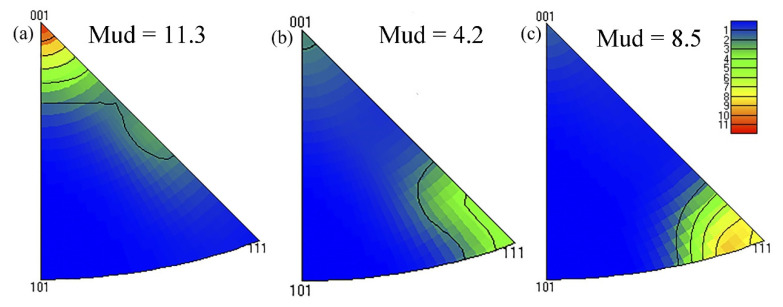
Inverse pole figure of alloys 1 (**a**), 2 (**b**), and 3 (**c**) after aging treatment.

**Figure 8 materials-15-00443-f008:**
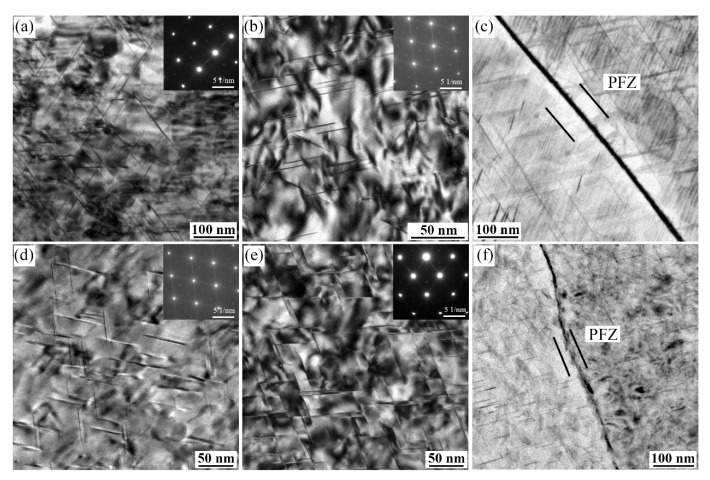
Transmission electron microscopy (TEM) images of alloys 1 (**a**–**c**) and 3 (**d**–**f**) aged at 160 °C for 32 h in the T6 temper (**a**,**c**,**d**,**f**) and T8 temper (**b**,**e**) taken along the [110]_Al_ direction.

**Figure 9 materials-15-00443-f009:**
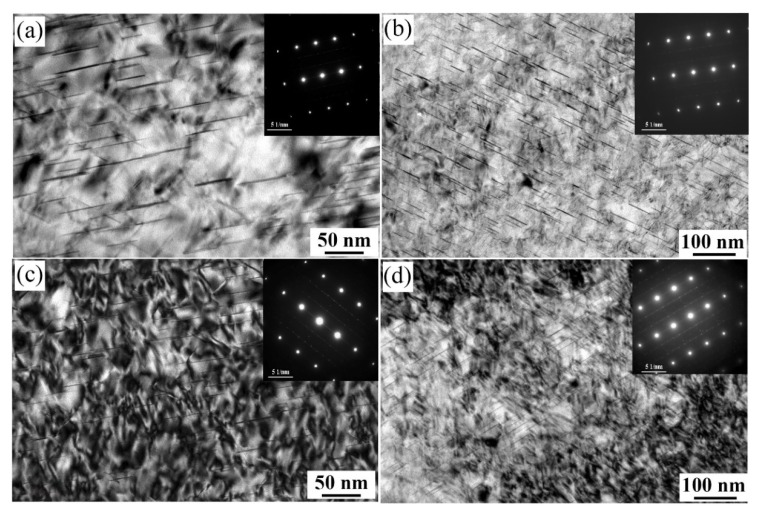
Transmission electron microscopy (TEM) images of alloys 1 (**a**,**b**) and 3 (**c**,**d**) aged in the T8 state for different aging times taken along the [112]_Al_ zone axis: (**a**,**c**) 32 h; (**b**,**d**) 80 h.

**Figure 10 materials-15-00443-f010:**
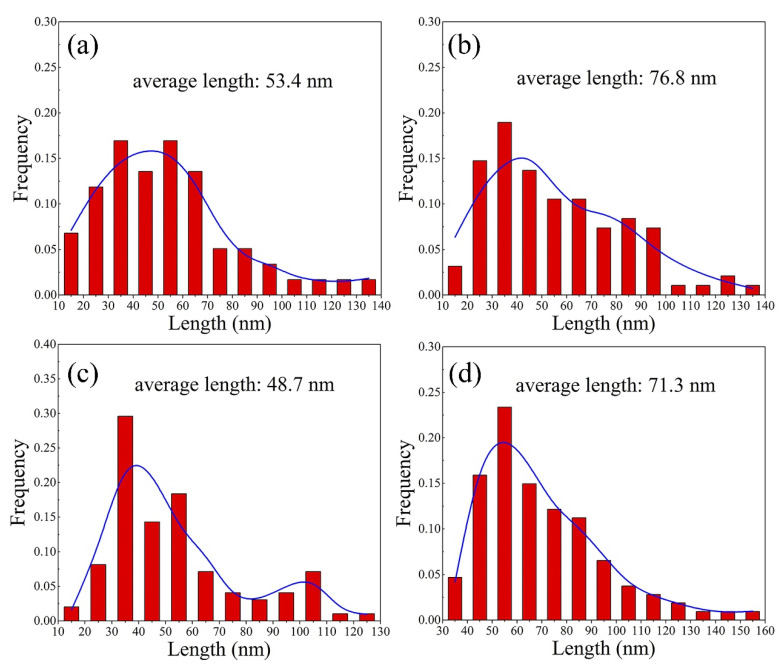
The length distribution of T1 phases from alloys 1 (**a**,**b**) and 3 (**c**,**d**) aged in the T8 state at different aging times: (**a**,**b**) 32 h; (**b**,**d**) 80 h.

**Figure 11 materials-15-00443-f011:**
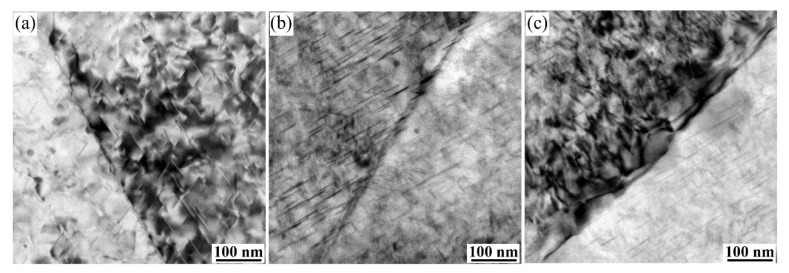
Transmission electron microscopy (TEM) images showing the distribution of precipitates from alloys 1 (**a**) and 3 (**b**,**c**) in the subgrain boundaries.

**Figure 12 materials-15-00443-f012:**
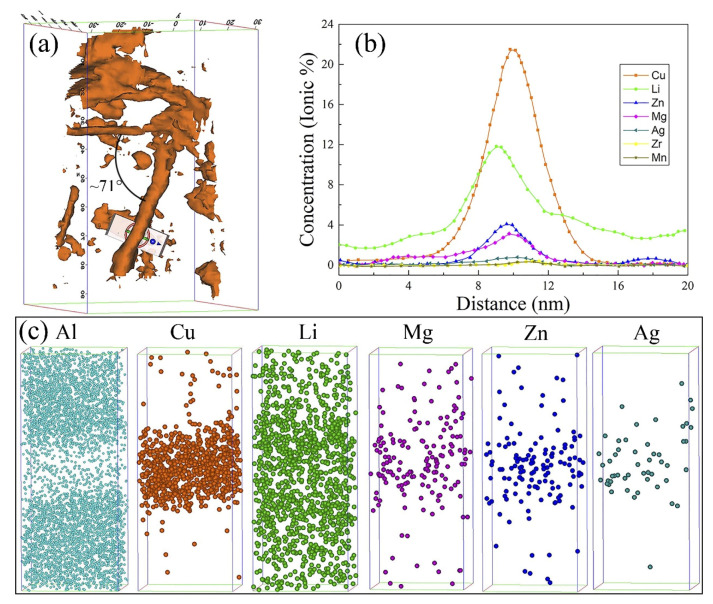
Atom probe tomography (APT) diagram of the alloy 3 aged at 160 °C for 32 h in the T8 temper: (**a**) perspective view, the orange is 11 at.% Cu iso-concentration surfaces; (**b**) concentration profiles through the plate in (**a**); (**c**) Al, Cu, Li, Mg, Zn, and Ag atom maps.

**Table 1 materials-15-00443-t001:** The actual chemical composition of alloys.

Samples	Element/wt.%								Density /g·cm^−3^
	Cu	Li	Mg	Ag	Mn	Zr	Zn	Al	
Alloy 1	3.90	1.10	0.60	0.30	0.20	0.15	--	Bal.	2.67
Alloy 2	3.97	1.17	0.51	0.27	0.21	0.11	0.29	Bal.	2.67
Alloy 3	3.99	1.07	0.60	0.34	0.31	0.13	0.57	Bal.	2.69

**Table 2 materials-15-00443-t002:** Electrochemical parameters of alloys under different aging conditions.

Aging Condition	Alloy 1		Alloy 2		Alloy 3	
	E_corr_ (V_SCE_)	I_corr_ (A/cm^2^)	E_corr_ (V_SCE_)	I_corr_ (A/cm^2^)	E_corr_ (V_SCE_)	I_corr_ (A/cm^2^)
T6	−0.665	3.92 × 10^−6^	−0.685	2.03 × 10^−6^	−0.697	1.78 × 10^−6^
T8	−0.708	1.10 × 10^−5^	−0.711	3.17 × 10^−6^	−0.727	1.81 × 10^−6^

## Data Availability

The raw/processed data required to reproduce these findings cannot be shared at this time as the data also form part of an ongoing study.

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
