# Peer review of "Microstructure, Mechanical Properties and Corrosion Behaviors of Al–Li–Cu–Mg–Ag–Zn Alloys"

_materials, 2022, doi:10.3390/ma15020443_

Round 1
Reviewer 1 Report
The paper presents a study on the effects of Zn content (in two concentrations - 0.29 and 0.57 wt.%, compared to a reference) with an in-depth study the structure, mechanical properties and corrosion resistance in Al-Li-Cu-Mg-Ag-Zn alloys. Such Al-Cu-Li-based alloys are important in high-priority structural applications, especially in aerospace and automotive industry, where a combination of mechanical strength and corrosion resistance is crucial, hence the study of the effects of solute additions in the improvement of the above mentioned properties is important. Due to this I believe that the topic will be of interest to the readers of Materials and can recommend it for publication, after a few minor improvements.
All in all, I am very pleased with the proposed manuscript. While the number of compositions, which are studied, is relatively minor (objectively one control composition and two different concentrations), the authors have demonstrated a commendable rigour in the details of their characterisation with multiple methods and discussion of the results.
The English presentation is also on a commendably-high level. There are some minor issues with miss expression and in most cases the improper use of plural (e.g. in the final paragraph of Introduction “mechanical properties, corrosion behaviors, and ….. mechanism of Zn on crystal structure and the electrochemical behaviors”, which I am certain will easily be corrected during the copy editing stage.
One of the major issues is that the manuscript is not using the MDPI manuscript template, and the referencing style does not completely comply with the author's guidelines, however, based on the quality of the work I suggest that it is recommended for publication.
Several minor notes:
1) In the Introduction the term “free precipitation zone” is abbreviated as PFZ, maybe it should be altered to the more correct term precipitate-free zone in order to correspond to the abbreviation.
2) Item 2 in the manuscript is titled “2. Experiences”, which is probably a mistake. Please correct.
3) Under the experimental section the style is similar to a letter. I suggest that this section is split into several sub-sections: e.g. Specimen preparation, Mechanical properties determination, Structural and chemical composition, Corrosion and electrochemical studies in order to aid the reader, when referring to this section.
4) The authors should carefully spellcheck the bulk of the text - in some places there is inconsistency in abbreviations (e.g., ICG and IGC, where the latter is correct).
5) The atomic probe tomography method and procedure is not defined in the Experimental section.
6) While is clear that the manuscript is directed to a very specialised audience, familiar with alloy research, the authors may consider to add an introduction to T temper codes (T1 - T8) earlier in the text. There are a lot of references to them (including in the abstract on several instances).
Reviewer 2 Report
1)«The as-cast ingots were subjected to continuous three-stage homogenization treatment. They finally held at 500 ℃ for 24 h and then cooled in the furnace. Subsequently, the homogenized ingots were hot extruded into plates when heated at 450 ℃ for 80 minutes; the extrusion ratio of plates is 9. After the extrusion process, the extruded plates were immediately cooled by water. The alloys were solution-treated at 530 ℃ for 1h in molten salt and then quenched in water at room temperature.»
PLEASE describe in more detail the temperatures and purpose of the three-stage homogenization. What are the critical temperatures in the studied alloys? Why did homogenization take place at 500, and sst at 530?
Reviewer 3 Report
Drafted manuscript “Microstructure, mechanical properties and corrosion behaviors of Al-Li-Cu-Mg-Ag-Zn alloys” is interesting, well written. The results are discussed in detail. Some minor comments should be improved.
- Abstract.
“ The Zn-added alloys contain a great number of non-recrystallized grains and a few fines recrystallized grains, while the T1 phase with finer dimension and higher number density, could explain the constantly increasing tensile strength. In addition, increasing Zn contents led to a lower corrosion current density and a shallower maximum intergranular corrosion depth, thus improving the corrosion resistance of the alloys.”
“great number” and “few” are not scientific words. Please provide quantitative results.
- Introduction
“Besides, Zn addition can also improve the mechanical properties of Al-Li-Cu alloys [17, 18]. Sun et al et[19] discovered that Zn efficiently enhanced the precipitation of T1 phases while inhibited the formation of θ' phases in the matrix.”
“θ' phases” – one phase. Not a phases.
- “Experiences” or “Experimental”?
Please provide a description of the thermomechanical treatment. Homogenization – 500C, hot extrusion – 450C and solution treatment – 530C.
- Conclusions look like a technical thesis. Please add a description “why”.
Reviewer 4 Report
The article is properly prepared, the latest methods are used, and the latest literature has been studied. For a comprehensive description of the alloy, I miss in more detail visualization by scanning electron microscopy for analysis of precipitations.
I suggest the following corrections:
- The method designations below the images are missing (light microscopy, SEM, EBSD, BT-TEM, SAED….).
- In this case, the characterization with light and partly transmission microscopy (the STEM and HRTEM is missing) is sufficient, but the exact characterization (higher magnifications) with SEM is missing.
I suggest you add the characterization the distributions and the shape of the precipitates with SEM, which has a larger field of view and depth of field compared to TEM.
- The description of the device is not precise enough: HVS-50 microhardness instrument
the correct record of hardness on graphs (HV3) is also missing (fig. 1)
